**REVIEW**                                                                                                   **Open Access**

# Navigating the crowd: visualizing coordination between genome dynamics, structure, and transcription

Haitham A. Shaban[1,2*] , Roman Barth[3] and Kerstin Bystricky[4,5*]

* Correspondence: h_shaban@
aucegpt.edu; kerstin.bystricky@univ-
tlse3.fr
[1]Spectroscopy Department, Physics
Division, National Research Centre,
Dokki, Cairo 12622, Egypt
[4]Laboratoire de Biologie Moléculaire
Eucaryote (LBME), Centre de
Biologie Intégrative (CBI), CNRS,
UPS, University of Toulouse, 31062
Toulouse, France
Full list of author information is
available at the end of the article

## Abstract

The eukaryotic genome is hierarchically structured yet highly dynamic. Regulating transcription in this environment demands a high level of coordination to permit many proteins to interact with chromatin fiber at appropriate sites in a timely manner. We describe how recent advances in quantitative imaging techniques overcome caveats of sequencing-based methods (Hi-C and related) by enabling direct visualization of transcription factors and chromatin at high resolution, from single genes to the whole nucleus. We discuss the contribution of fluorescence imaging to deciphering the principles underlying this coordination within the crowded nuclear space in living cells and discuss challenges ahead.

## Introduction

A largely unexplored question in chromatin biology is how chromatin organization and dynamics relate to function, a question that needs to be addressed in single living cells and at multiple spatio-temporal scales to be answered. DNA transcription, in particular, is a vital but sensitive process to which many players contribute in order to define and maintain cellular identity [1]. These players act through promoter and enhancer sequence elements of one or even several genes. Finding the right partner in a highly crowded environment is a tedious task [2, 3], because multiple enhancers are frequently distant from the promoter of the gene to be regulated along the one-dimensional genome. Furthermore, the motion of chromatin, intrinsically constrained by the mere length of the chain, is hampered in crowded environments [4], making exploration of space a slow process [5]. Recent advances in molecular biology, imaging, and physical and mathematical modeling have improved our understanding of how transcription depends on the nuclear organization and how this organization may facilitate transcription.

Chromatin conformation capture methods [6, 7] are now routinely used to interrogate the chromatin structure of cells in many organisms and in response to various stimuli. However, all sequencing-based methods are by nature invasive and thus make

microscopy-based single-cell time course experiments a necessity to understand the dynamic behavior of chromatin. For a comprehensive comparison between different imaging techniques used to studying genome organization and transcription, we refer the reader to recent reviews [8–11]. In addition, imaging techniques can be multiplexed for simultaneous visualization of a multitude of chromatin constituents and are amenable to high-throughput analysis of genome organization at kilobase resolution [10]. While super resolution imaging-based approaches are often performed in fixed cells, time-resolved whole-chromatin imaging recently revealed that chromatin moves in a spatially and temporally correlated manner at a nanoscale resolution [12, 13]. Such correlated motion might be caused by active mechanisms [14], including transcription [15, 16] and/or polymer properties [17]. In silico, physical modeling of data from chromosome conformation capture techniques and fluorescence imaging produced several models which describe genome organization in the context of biological processes such as transcription [18, 19]. Comprehensive time-resolved data complemented with static and ensemble data now bear great promise to correlate changes in the 3D organization to transcription.

Here, we describe how recent advances in live-cell imaging of chromatin in eukaryotic cells enriched our understanding of genome organization at different spatial and temporal scales. We start by considering the complementary views that well-established and yet continuously developing sequencing-based "C"-methods (henceforth Hi-C) and imaging-based methods can offer. We then outline various strategies to interrogate chromatin structure and/or dynamics in three-dimensional space and time, from a single gene to the complete genome, and describe how transcription activity influences chromatin folding and dynamics. We finally discuss how physical principles can explain the spatio-temporal coordination between the chromatin structure and dynamics and transcription in the nucleus.

## Fluorescence imaging and Hi-C: complementary and controversial views

Hi-C revolutionized our knowledge of genome structure: a hierarchy of structural elements was discovered, the most prominent ones being DNA loops, topologically associated domains (TADs), and compartments [20–22]. While those elements are derived from a population average of structures, single-cell Hi-C has confirmed what cell biologists have known for decades that chromosome and chromosome domain conformations vary from cell to cell [23–27]. Distance measurements between FISH-labeled loci can validate selected features seen in models established from Hi-C maps [28–30], but distances do not necessarily correlate with contact frequencies [27, 31, 32], a tendency confirmed by high-throughput approaches [33, 34]. Furthermore, it is tedious to probe distances throughout the whole genome since FISH probes must be designed for each region of interest and can thus detect only a subset of chosen genomic loci in a reasonable time.

Hi-C and derived methods can assess the probability of crosslinking which is a measure of the frequency that chromatin loci are in proximity, probing spatial distances in the range of ~ 400–600 nm [35]. By imaging, it is possible to quantify the distribution of actual spatial distances between any two or more loci (Fig. 1), making "C" and imaging complementary approaches [37]. Imaging also allows to characterize the shape of a given genomic region of interest [38] and to monitor the positioning of genes relative to nuclear compartments which can be visualized simultaneously [33, 39, 40] (Fig. 1).

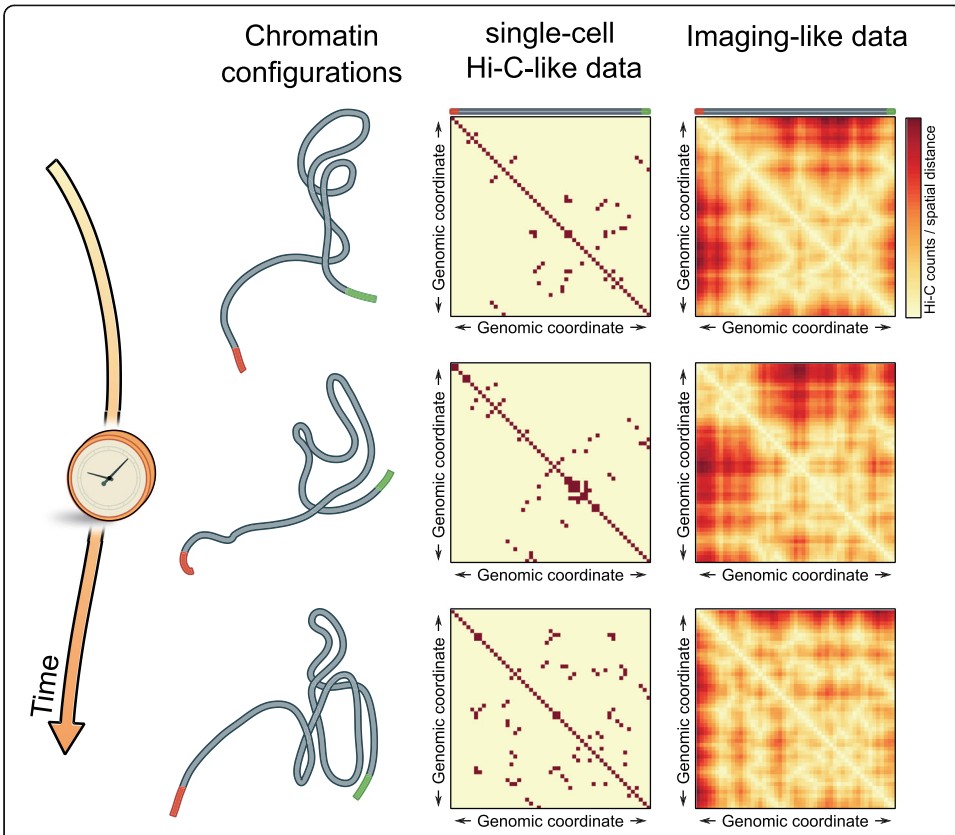

**Fig. 1** Advances in imaging techniques allow rich information about chromatin beyond Hi-C. Chromatin is constantly remodeled during time (top to bottom), which is illustrated on the basis of a short polymer which changes its configuration over time (left). A single-cell Hi-C-like type of data over time of this polymer would reveal relatively few contacts at each time point for single cells as the technique relies on proximity ligation (middle column). In contrast, imaging offers the determination of actual positions and distances between any two loci in three dimensions and thus reveals a more complete picture than "C" methods (right column). The illustrative maps were created by tracing the contour of the polymer shown on the left and computing the pairwise distance between any two loci which is shown in the imaging-like matrix. The Hi-C map is a thresholded version of the distance map and shows contacts only at small spatial distances. Yet, note that there may be a broad distance distribution underlying measured Hi-C contacts [36], and as such, the illustration is highly simplified. The Hi-C map is a thresholded version of the distance map and shows contacts only at small spatial distances. While imaging chromatin at many loci simultaneously is currently, with a few exceptions, done in fixed cells, it has the potential to advance toward analysis in living cells in the future. However, a single-cell time evolution of chromatin structure by Hi-C cannot be obtained since Hi-C is a destructive method

With the advent of sequential FISH [31], the three-dimensional folding of chromatin regions was probed directly in single cells and in situ (Fig. 2). Cooperative three-way interactions between loci [41] not seen previously using population-averaged contact maps were identified. Imaging further showed that TADs appear as discrete nano-compartments, which are spatially arranged side by side [41–43], although contact frequencies between loci within TADs only slightly (~ 2-fold) increased compared to loci within neighboring TADs [27, 44]. Integrating multiplexed imaging of RNA [45, 46] with sequential DNA FISH [42, 43] revealed that genes located at TAD boundaries were transcribed more frequently than genes within the TAD's interior highlighting that features of local chromatin structure and transcription activity are related.

Despite their success, three-dimensional chromatin structures derived from either sequential DNA FISH imaging or reconstructed from Hi-C maps are currently limited to

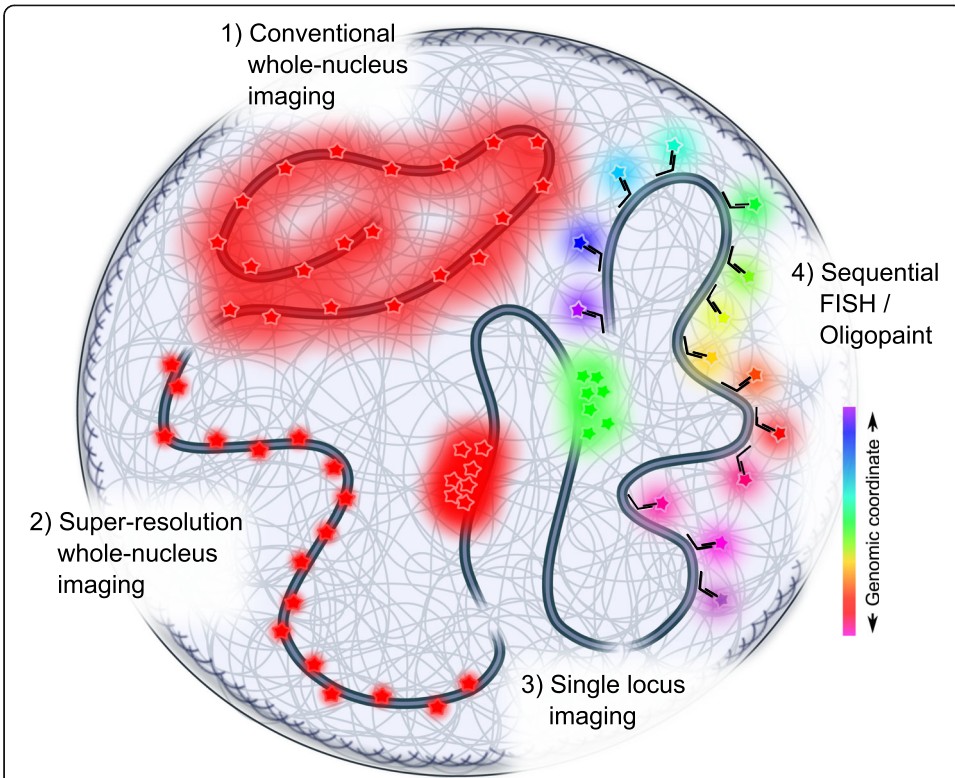

**Fig. 2** Labeling and imaging strategies to image chromatin. Conventional labeling using stably expressed fluorescent proteins or organic dyes usually covers the whole genome unspecifically (modality 1). Due to the high density of chromatin, only a spatial resolution well above the diffraction limit can be achieved, while fast imaging is generally possible. Super-resolution imaging of chromatin overcomes the diffraction limit but is challenging since chromatin in vivo is not well structured (modality 2). Usually, resolution is gained for the expense of acquisition time and thus cells must be fixed. While conventional and super-resolution whole-genome labeling is sequence-unspecific, single loci can be targeted using fluorescence in situ hybridization (FISH; in case of fixed cell imaging), ANCHOR, a CRISPR/Cas9 system, or others (modality 3). Single locus labeling can be done in fixed as well as in living cells. To cover an extended region of chromatin for which sequence information is available, a sequential FISH or oligopaint strategy can be followed in fixed cells (modality 4). Using oligonucleotides which are designed for the specific genomic region, the chromatin configuration is sampled by sequential hybridization rounds and computationally reconstructed

(i) fixed cells and (ii) a genomic resolution of ~ 2 kb [43], although sequencing-based techniques to map contacts at the nucleosome level are being developed [47, 48]. As an alternative approach, sequence-unspecific super-resolution imaging of histone H2B and of histone post-translational modifications was employed to visualize chromatin in fixed [49, 50], and recently also in living, cells [12, 51] (Fig. 2). These approaches revealed that nucleosomes transiently associate in groups. Since the genome and associated processes are inherently dynamic and many DNA-proteins as well as DNA-DNA contacts are transient, fixed cell studies naturally cannot capture this process and population assays represent a blurred picture over many cells. Time-resolved imaging is thus needed to appreciate the dynamic nature of the biological processes such as transcription in single cells. However, it is also evident that imaging specific sequences is necessary to place time-course data into a functional context. In the future, labeling strategies will need to be designed for fluorescence live-cell imaging to simultaneously

observe structural changes of chromatin and its functional consequences simultaneously in the same cell (Fig. 1).

## Chromatin dynamics from single genes to the entire nucleus

### Motion of individual chromatin loci is subdiffusive in living eukaryotic cells

To gain insights into the dynamic behavior and spatio-temporal organization of chromosomes in live cells, distinct genomic sites can be labeled and tracked in real time (Fig. 2). Several methods have been developed to create a fluorescent signal distinguishable from background fluorescence [52]. Labeled loci can then be tracked over various length and time scales using single-particle tracking (SPT) approaches [53]. Gene editing tools contributed largely to visualize chromatin. The first method to tag chromatin loci and monitor their motion in real time used a chimeric DNA binding lac repressor - GFP fusion protein targeted to multiple repeats of operator binding sites [54, 55]. This approach showed that chromatin motion is constrained and anomalous and influenced by nuclear structures, cell cycle, and function [55–57]. A more innocuous system suited to study changes in chromatin motion in close vicinity of a genomic locus without disrupting its function is the ANCHOR/ParB DNA-labeling system. ANCHOR implants a short non-repetitive sequence near the gene of interest which triggers the accumulation of fluorescent ParB proteins [58, 59]. Several editing-free systems such as transcription activator-like effectors (TALE) [60, 61] and clustered regularly interspaced short palindromic repeats (CRISPR/inactive dCas9) [62, 63] have also been used to visualize genomic loci [62–64]. These systems frequently suffer from poor SNR, but tracking of repetitive DNA sequences which provide a large number of sites confirmed that their motion is also constrained [62, 63, 65]. Most of these SPT studies concluded that the majority of mammalian interphase chromatin motion is subdiffusive [59, 66–68] and in some cases superdiffusive [69, 70].

### Motion of individual chromatin loci near genes changes in response to transcription activation in living eukaryotic cells

The prevalence of inactive genes to localize near the nuclear periphery and active genes to prefer the nuclear interior suggests that those loci undergo long-range movements upon gene activation toward the nuclear interior [69]. This can however occur over hours or days, during development and differentiation, and cannot inform on the underlying physical properties of the chromatin fiber (for review see [71]). To determine how activating transcription influences the local motion of chromatin, a few recent studies have examined the behavior of tags inserted in the vicinity of genes. Combining the ANCHOR DNA-labeling system with MS2-labeled mRNAs enabled monitoring the motion of a single transgene under control of a hormone inducible endogenous promoter before and after transcription in the same cell [59]. While the motion of the single-gene domain remained subdiffusive, transcription initiation caused local confinement of the gene within seconds only when mRNA was produced in the analyzed cells [59]. Confinement was maintained during elongation by RNA polymerase II (RNAP II). These findings suggest that increased chromatin collisions and assembly of functional protein-DNA complexes during the steps of transcription reduce motion, coherent with an increase in local crowding (discussed below). An imaging system

based on chimeric arrays of gRNA oligonucleotides (CARGO) to which dCas9-GFP fusion proteins bind was applied to monitor the *Fgf5* enhancer and promotor loci in mouse embryonic stem cells (ESC) before and 4 days after inducing cellular differentiation [64]. In contrast to the study in mammary tumor cells by Germier et al. [59], the study by Gu et al. [64] reported that mobility increases at the enhancer and promoter of the target gene when its activation was inhibited [64]. The increase in calculated diffusion coefficients is explained by non-thermal agitation of chromatin domains, which may favor enhancer-promoter contacts as a result of the stochastic confrontations within the TADs [64]. However, the establishment of a TAD as defined by ensemble approaches does not necessarily insulate from inter-TAD contacts in single cells and is thus unlikely to promote enhanced contact frequencies upon enhanced dynamics. Contacts (within a physical distance of < 10 nm) of two loci can thus only be reliably enhanced at loop bases due to the repeated extrusion of the region governing the loci of interest into loops, not because TAD formation promotes contacts of two loci within TADs at any time. It is also possible that changes in global chromatin organization in ES cells before and after 4 days of inducing differentiation have different dynamic properties independently of the motion of any individual gene or regulatory locus. This has yet to be tested. While recent studies of chromatin dynamic imaging at single nucleosome and global chromatin levels support the finding that chromatin is confined in transcriptionally active nuclei compared to cells treated with RNAP II inhibitors [16, 72], studying single gene motion in different cell lines (e.g., human cancer cells vs mouse ESC) using the same experimental setup and analysis method would be important.

### Chromatin dynamics are spatially partitioned in nuclear domains

Early tools for visualizing chromatin within the entire nucleus either using the expression of fluorescent histones or by incorporating transfected fluorescent nucleotides during replication [73, 74] contributed later for studying the global dynamics of the genome. The alliance between photo-activated localization microscopy (PALM) and SPT is a powerful tool for quantifying chromatin dynamics at the single nucleosome level for sparse domains [16, 75]. Relying on the photoactivation of PAFP fluorophore-tagged histone H2B, this method tracks unbleached nucleosomes for up to 500 ms [75]. However, longer acquisition time is required for accurate analysis and in-depth understanding of the processes governing nucleosome dynamics at the physically and biologically relevant time scales. New organic fluorophore tags of DNA such as Hoechst 33342 and its far-red variants (silicon-rhodamine-Hoechst) [76, 77] have been introduced to circumvent phototoxicity and photobleaching that enabled analyzing the entire chromatin in vivo for up to 30 s [15, 72].

Imaging abundant nuclear (macro-) molecules such as chromatin, transcription factors (TF), or RNA poses limitations on the application of common methods such as SPT to analyze their dynamics. Particle identification in a densely labeled environment is not always unique and may even be impossible due to the limited spatial resolution. As a complement to SPT for studying chromatin motion, raster image correlation spectroscopy, pair correlation analysis, and optical flow-based methods have been applied to quantify global chromatin motion in living cells [15, 67, 78, 79]. The high-resolution diffusion mapping (Hi-D) method reconstructs trajectories at nanoscale precision

across the entire nucleus simultaneously [72]. A Bayesian inference approach is used to infer types of diffusion for every single pixel. Consequently, the resulting high-resolution maps of global chromatin diffusion provide biophysical parameters (type of diffusion, diffusion constant, anomalous exponent) of different nuclear domains in single living cells [72]. Chromatin domains spanning 0.3−3 μm positioned in a mosaic-like manner were detected in the nuclear interior. These domains were seen to be remodeled in response to transcriptional activity. Strikingly, the dynamic properties of these patterns were uncoupled from chromatin density [72]. Emerging whole-chromatin super-resolution techniques could detect nanodomains, consisting of up to ten nucleosomes, as a small-scale structural entity (diameter < 100 nm) of the genome [49, 51] and further developments enabled to determine whole-chromatin chromatin dynamics well below the diffraction limit to link structural changes of these entities to dynamics [12]. Mapping nuclear dynamics of dense molecules across the entire nucleus simultaneously instead of the isolated visualization of single loci opens new perspectives for our understanding of genome motion in the context of nuclear architecture.

## Spatio-temporal organization of transcriptional dynamics

Transcription is a highly regulated and dynamic process [80] and is set up by transcription initiation controlled by binding and release of TFs to DNA sequences and transcriptional machineries [81, 82]. TF-binding kinetics and diffusion within the target DNA domains in live cells were originally determined using fluorescence recovery after photobleaching (FRAP) [81, 83]. Later, improving the signal to noise ratio (SNR) of labeled molecules enabled detecting and quantifying binding events of TFs to DNA in two [84] and three dimensions [85]. Fluorescence microscopy methods such as FRAP, fluorescence correlation spectroscopy (FCS) [86], SPT, and super-resolution microscopy have greatly contributed to uncover and quantify the variability of TF dynamics [87], including interactions with their genomic targets by measuring the residence times, on/off rates, and diffusion constants [80, 88]. The majority of these studies showed that the residence time of TFs at specific and non-specific binding sites varied by some orders of magnitude, with longer residence time for specific binding (milliseconds to a few seconds) than for non-specific binding (tens to hundreds of milliseconds) [84, 89]. The residence time of bound TFs may alter the spatio-temporal expression patterns, dynamics, and nuclear organization of TF-loaded loci in live cells [90]. Specifically, swift TF release is required for regulating expression patterns. Nonetheless, long residence times were recorded at some sites, making the process less tunable [80, 91, 92]. Long-time imaging of the TF Sox2 and chromatin domains recorded jumps of Sox2 between binding sites suggesting that kinetic chromatin domain structures facilitate transcription regulation at this locus in embryonic stem cell [87, 93]. Yet, contacts between Sox2 and its enhancer, if any, were too short lived to be imaged [94] in an engineered system. Simultaneous imaging of labeled promoter, enhancer, and/or gene sequences in cells in which the local transcriptional state is known could in the future unequivocally characterize relative positions, possible contacts, and their frequencies between these transcriptional elements and sharpen our understanding of their coordination from transcription initiation to termination.

Transcription is not only regulated in time but depends largely also on the spatial organization of TFs and RNAP II [95]. Furthermore, transcription can modify the architecture and dynamics of DNA [96]. Nuclear gene positioning, packing, and looping

of transcriptionally active loci are dynamic, granting proteins access to regulate the transcription processes at different levels [2, 97–99]. Likewise, the spatial distribution of TFs and RNAP II is highly imparted upon transcription initiation; for instance, RNAP II was found to cluster in transcription factories in early fluorescence and electron microscopy studies [100–104]. These transcription factories may form anchor points to which chromatin is tethered, thereby co-regulating the dynamics of chromatin and proteins that compose the factories [16]. Indeed, the motion of global chromatin and RNAP II correlates when examined at a nanoscale dynamic resolution and a time resolution of 5 fps [72], and chromatin dynamics are locally [59] and globally [16, 72] confined upon transcription. Furthermore, the formation of correlated chromatin domains could be linked to transcriptional activity. The correlated motion was decreased but not lost completely by blocking RNAP II activity [15, 67]. These results indicate that stopping transcription at early stages does not eliminate long-range contacts and does not dissolve transcription factories [105, 106]. Spot tracking analysis of the even-skipped (eve) locus promotor and enhancer during transcription in live *Drosophila* embryos identified three topological states: an open state which is however inactively transcribed and a paired *homie–homie* state which can be both actively and inactively transcribed [107]. Accumulation of nascent mRNA indicates that transcription initiation transiently enhances the stability of the proximal configuration between enhancer and promoter and increases the gene's spatial compaction, consistent with hindered motion [59, 107, 108]. These examples highlight the role of transcription in shaping the hierarchical organization of mammalian genomes in the nuclear space as well as their dynamics.

## Mechanisms governing the dynamics of the genome during transcription

Several mechanisms and models were proposed to explain the nature of the observed chromatin dynamics. Here, we discuss the most popular and experimentally—by fluorescence microscopy and biochemical methods—well-established principles.

### Liquid-liquid phase separation as a physical model to understand the regulation of gene expression

Upon transcription initiation, hundreds of proteins have to reach a transcription start site in a highly coordinated manner [109]. Live-cell super-resolution imaging revealed that RNAP II engages transiently in transcription factories during transcription initiation [110, 111] (Fig. 3 (A)). The mechanism by which transcription factories form could be resolved by lattice light-sheet imaging in living cells: transcription factories behave like liquid condensates [112], membrane-less compartments, likely formed spontaneously by a liquid-liquid phase separation (LLPS) mechanism due to spatial concentration heterogeneities of the condensate components [113]. These findings found support [112, 114–119] in that the carboxyl-terminal domain (CTD) of RNAP II, an intrinsically disordered low-complexity region, can undergo cooperative LLPS in vitro [117]. The condensates are dissolved upon CTD phosphorylation [117], which is the same mechanism by which RNAP II is braced for transcription elongation [118, 120]. By controlling CTD phosphorylation, the contact duration between transcription factories and DNA may thus be regulated [110, 121]. This mechanism appears to be common to several other transcription factors in vivo [119],

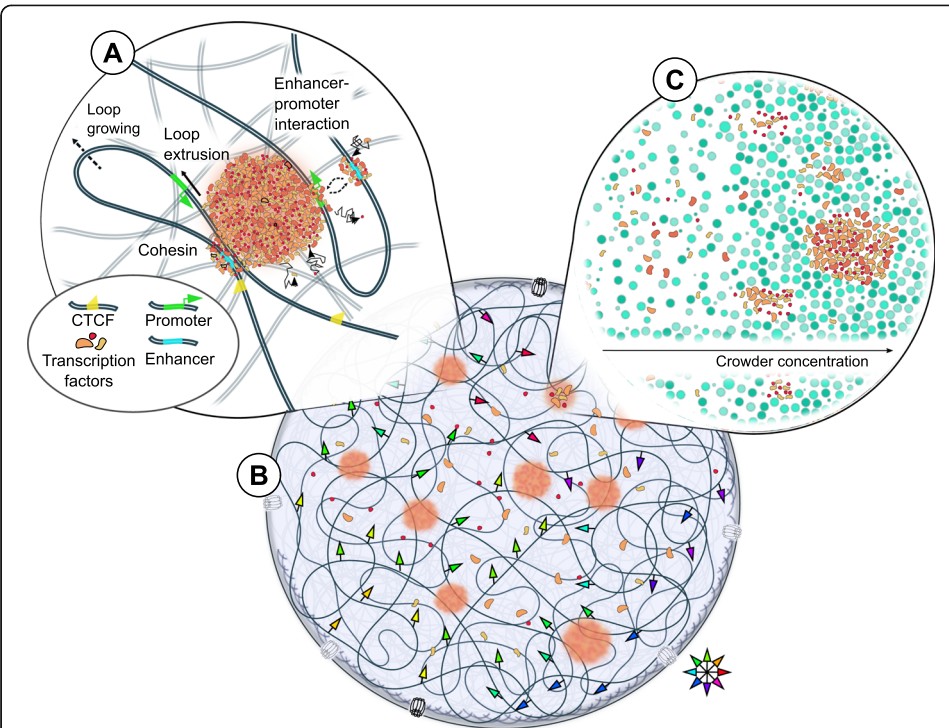

**Fig. 3** Mechanisms for chromatin organization and dynamics during transcription. (A) The transcriptional hubs are formed by liquid-liquid phase separation of transcription factors, which in turn is mediated by the local crowding conditions. TFs binding to both enhancer and promoter mediate enhancer-promoter contacts by an effective attraction exerted by the LLPS property of transcription factories. Once transcription is initiated by these enhancer-promoter contacts, transcription elongation proceeds by reeling the transcribed gene body along the transcription factory. Loop extrusion by cohesin can additionally establish transient enhancer-promoter contacts, and the turnover time of cohesin and CTCF regulates the frequency of these contacts. Loop extrusion dynamics and the placement of (semi-) permeable border elements may thus regulate transcription. (B) The nucleus is sprinkled with transcription factories to which chromatin is tethered. The resulting network of transcriptional hubs restricts chromatin motion and induces a stiffness to chromatin, which is expressed in long-range correlations of chromatin dynamics (colored arrows). (C) The local molecular crowding reduction of chromatin mobility upon transcription. This transition is associated to high molecular crowding, eventually facilitating the formation of transcription factories to which chromatin is tethered

which led to the current view that transcription factories could be condensates in vivo, whose intrinsically disordered low-complexity sequence domains undergo liquid-liquid phase separation (LLPS) (Fig. 3 (A, B)).

Super-enhancers, clusters of putative enhancers in close proximity with unordinary high density of transcriptional co-activators binding [112, 115, 116], have shown the potential to form a nucleation point for LLPS, directed by master TFs [122]. The unusually high transcription activity of super-enhancers has to be "contained" within insulated neighborhoods, demarcated by Hi-C-defined TAD boundaries, in order to specifically activate their target gene [122]. As such, the local chromatin architecture appears to be vital to prevent ectopic gene activation.

LLPS does not only concern protein condensates such as transcription factories but may also interact with and actively shape the surrounding chromatin environment [123] (Fig. 3 (B)), in particular, at super-enhancers [124, 125]. For instance, the fusion of condensates minimizes their surface tension and deforms chromatin locally which may bring chromatin loci in the vicinity of each other [123]. Merging of condensates of

TFs at promoters and enhancers could explain how these potentially distant genomic regions find each other (Fig. 3 (A)). Tethering of chromatin to transcriptional hubs could also account for reduced freedom of chromatin movement [16], and result in a stiffer DNA network, which is detected by spatially coherent motion [15, 67] (Fig. 3 (B)). Furthermore, chromatin itself can self-organize and phase-separate, consistent with entropic forces of polymers [126]. For instance, heterochromatin is preferentially found at the nuclear periphery and near nucleoli, and repressed genes are found buried inside nuclear compartments, while active genes and RNA-associated proteins preferentially lie on the surface of chromatin domains [51, 127]. Fluorescence microscopy is an excellent tool to detect and quantitatively study such LLPS-driven chromatin bodies and compartments [128].

### Molecular crowding drives chromatin dynamics, confinement, and function

Molecular crowding significantly influences the diffusion of proteins within the nucleus and is a driver for transcriptional silencing within heterochromatin by phase separation [4, 129, 130]. Crowding affects the binding affinity of RNAP II on DNA [131, 132] such that in vitro transcription is markedly enhanced only when crowder volume fractions resemble those found in vivo [133, 134]. Especially temporal changes in crowder density, as opposed to a commonly assumed steady-state crowder concentration, have been shown to be able to selectively up- or downregulate genes and could thus constitute an additional gene regulatory pathway for the cell [135]. Because the microenvironment of a chromatin locus relates to its dynamics [136–138], such crowding density fluctuations may be captured by varying chromatin dynamics over time. The reduction of chromatin mobility upon transcription suggests that this transition is related to a local accumulation of crowders, eventually facilitating phase separation of transcription factors to which chromatin is tethered [16, 72] (Fig. 3 (C)). Finally, it should be noted that not only proteins, but also chromatin itself, is a crowding agent. The observation that LLPS takes place in regions of low chromatin occupancy [123] could relate to the fact the local crowder concentrations can control the occurrence of LLPS. It is likely that several physical forces resulting from macromolecular crowding, chromatin compaction, and LLPS cooperate in order to spatially regulate transcription [139, 140].

### Coherent motion of chromatin depends on enzymatic activity

Chromatin was shown to move coherently over several microns by live-cell imaging. This coherent motion was modulated by activating DNA processes [15, 67] (Fig. 3 (B)). Computational models of this motion suggest that chromatin is dynamically partitioned [5, 17]. Intriguingly, boundaries of coherently moving domains appeared uncoupled from domains of similar chromatin compaction, but not from transcription states [15, 72]. Self-organizing properties of active systems, propelled by ATP-driven proteins and promoted by the long-scale correlation due to hydrodynamic interactions of the activity-induced motion, could be at the origin of coherence [14, 67, 141–143]. However, it should be pointed out that qualitatively similar findings can be yielded computationally when only chromatin chain conformation and epigenetic marks are taken into account [5, 17, 144]. Thus, the role of active chromatin remodelers remains to be further evaluated. Combining quantitative fluorescence imaging, Hi-C, and computational polymer modeling to study the mobility of yeast

chromosomes demonstrated how heterogeneous binding patterns of proteins along the chromatin fiber could lead to thermodynamically driven self-organization and differential mobility [145]. Thus, while transcription or other energy-consuming processes can induce coherent motion, additional players shaping chromatin architecture and epigenetic landscapes should not be overlooked when interpreting live-cell imaging data of chromatin.

### Loop extrusion can create windows of opportunity for fine-tuning transcription

The *transcription factory model* has been proposed to explain the mechanisms leading to genome folding [18, 93, 146, 147]. Binding of TFs to DNA induces a positive feedback loop by quickly re-binding dissolved TFs and recruiting new ones from the pool of freely diffusing TFs [18, 146] (Fig. 3 (A)). The resulting TF condensate is able to bridge different DNA regions and causes an effective bridging-induced attraction [146]. Merging of clusters at different DNA segments establishes chromatin loops, whose loop bases colocalize with TFs. This relatively simple mechanism was shown to be sufficient to recapitulate many features of chromatin structure established by Hi-C such as rosettes and TADs [148, 149]. This transcription factory model [18] supports the concept that transcription is an important driver of chromatin folding and dynamics [1, 15, 16, 72, 96], but also takes into consideration chromatin contacts mediated by proteins which are unrelated or only distantly related to transcription.

The *loop extrusion model* [150–152] conceptualizes how chromosome structure emerges due to the extrusion of DNA loops from structural maintenance of chromosomes (SMC) proteins [153–157] as loop extruding factors. Based on a collaboration between the SMC protein cohesin and molecular boundary elements (CCCTC-binding factor (CTCF) proteins), the loop extrusion model recapitulates the hierarchical organization of chromosomes revealed by Hi-C [158, 159]. Nevertheless, the model fails to reflect experimentally observed large-scale interactions [160], and it is now clear that phase separation mechanisms go hand in hand with loop extrusion [161].

In contrast, the transcription factory model poorly predicts details of interactions but correctly captures long-range contacts [160]. A combinatorial model consisting of the transcription factor and loop extrusion model faithfully reproduces many Hi-C features suggesting that TFs and cohesin have complementary roles in genome organization [160]. Chromatin domain boundaries are often found at long and highly expressed genes, whose strength is driven by transcription length and transcription rate [162], and most chromatin contacts are established between active transcription units [163, 164]. Furthermore, increased contact frequencies in *trans*, at the expense of *cis* contacts, as revealed by ChiA-PET [165], can be explained by the transcription factory model because this model is compatible with the idea that active genes on different chromosomes could be cotranscribed in the same transcription factory.

The loop extruding activity of cohesin was recently illustrated by in vitro experiments [157, 166, 167]. Given this mechanism, transient contacts between two genomic loci can be established, in particular enhancers and promoters, during the growth of the loop. The bulky transcriptional machinery is likely to influence the position of cohesin-mediated DNA loops [96, 168] (Fig. 3 (A)). However, how cohesin interacts with single or multiple stalled or elongating transcription complexes remains to be interrogated experimentally.

The loop extrusion and transcription factory models appear to complementarily explain how the genome is dynamically organized. Both models imply that transcription can influence the genome organization and vice versa.

## Future directions

Sophisticated labeling techniques as well as super-resolution microscopy enabled direct visualization of nuclear processes at several length scales, mostly well below the diffraction limit. By incorporation of data derived from orthogonal approaches, a more and more complete picture of chromatin structure (blob forming), dynamics, and function emerges. However, it must be kept in mind that the nucleus is a technically challenging organelle to analyze. Its content is highly crowded, and there are only few well-structured elements in the nucleus, which would facilitate the description of its interior by serving as reference points. Chromatin is a disordered, extraordinarily long, and confined biopolymer, which is constantly remodeled by a multitude of factors acting on it and that renders the analysis of chromatin and chromatin-associated processes highly complex.

Nevertheless, using advanced fluorescence imaging led us to a general picture in which transcription occurs in "transcription factories," condensates likely formed by liquid-liquid phase separation, to which chromatin is tethered and thus globally and locally constrained. Transcription appears in various aspects to be a key process to take into account in order to understand the dynamic chromatin landscape, even though other factors must also contribute [1].

However, the dependency between chromatin architecture and transcription regulation in vivo is a matter of debate. For instance, contacts between *cis*-regulatory elements precede gene activation during *Drosophila* development and remain regardless of the transcriptional state [169]. A few studies also found that enhancer-promoter proximity is not always indicative of the transcriptional state [94, 170]. Strikingly, auxin-inducible degradation of the insulator protein CTCF, which acts together with cohesin in human cells to shape chromatin loop domains/TADs, has only a mild impact on transcriptional deregulation [171], and auxin-induced degradation of cohesin has a similarly modest effect on transcription [172]. However, several lines of research demonstrated that the deletion of TAD boundaries can perturb transcription by interfering with ectopic enhancer-promoter contacts [173–175]. Genes which are deregulated upon loss of cohesin or CTCF are preferentially located at TAD boundaries, hinting toward a subtle regulatory role of factors shaping the genomic environment around *cis*-regulatory elements. Transient looping events by cohesin and/or TFs, which are likely not to be detectable by ensemble approaches such as Hi-C, could thus constitute a flexible mechanism by which genes can be both activated and repressed, for instance, as a response to stimuli.

To further elucidate how chromatin and transcription are regulated, four main challenges remain in the development of live-cell real-time imaging of nuclear processes and chromatin behavior: (i) sequence-specific labeling, (ii) multicolor imaging, (iii) super-resolution microscopy techniques to allow rapid imaging in three dimensions at a nanoscale resolution of densely distributed emitters, and (iv) improving computational tools that allow processing and visualization of a large pool of information in order to understand stochastic processes.

At the single locus level, labeling technologies for tagging of multiple non-repetitive sequences to target specific chromatin domains for live-cell imaging need to be further improved to link chromatin dynamics to function. Applying a lattice light-sheet [176] illumination would be a technical solution to enhance image acquisition speed and decreasing phototoxicity [177, 178]. In addition, machine learning algorithms offer a computational solution for processing and denoising of "low-quality" images acquired rapidly [12]. Multicolor live-cell imaging to directly visualize proteins and chromatin at the nanoscale and short time resolution will be a major feat.

## Supplementary Information

> **Additional file 1.** Review history.

**Peer review information**

Andrew Cosgrove and Kevin Pang were the primary editors of this article and managed its editorial process and peer review in collaboration with the rest of the editorial team.

**Review history**

The review history is available as Additional file 1.

**Authors' contributions**

H.A.S. conceptualized the project. H.A.S., R.B., and K.B. wrote the manuscript. All authors read and approved the final manuscript.

**Authors' information**

Twitter handles: @h_shaban (Haitham A. Shaban); @RomanBarth2 (Roman Barth); @kerstinbys (Kerstin Bystricky).

**Funding**

K.B. thanks the ANR Sinfonie grant for funding.

**Ethics approval and consent to participate**

Not applicable.

**Competing interests**

The authors declare that they have no competing interests.

**Author details**

[1]Spectroscopy Department, Physics Division, National Research Centre, Dokki, Cairo 12622, Egypt. [2]Current Address: Institute of Bioengineering, Ecole Polytechnique Fédérale de Lausanne (EPFL), Lausanne, Switzerland. [3]Department of Bionanoscience, Delft University of Technology, 2628 CJ Delft, The Netherlands. [4]Laboratoire de Biologie Moléculaire Eucaryote (LBME), Centre de Biologie Intégrative (CBI), CNRS, UPS, University of Toulouse, 31062 Toulouse, France. [5]Institut Universitaire de France (IUF), Paris, France.

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

## 