## [**Additional file 1.** Review history. · Genome Biology]

Review History

First round of review

Reviewer 1

Comments to author:

General comments:

The authors provide an overview of the state-of-the-art knowledge regarding chromatin dynamics in tridimensional nuclear space and relationship between genome organization and transcription.

The topic is at the forefront, the point of view innovative and certainly of great interest. Unfortunately, this reviewer had hard time to understand what the review is describing. Chapter are often not systematically organized, and sometimes partially cover the literature of the field. The main problem is that the causality between transcription and genome organization is still an unresolved issue, and the review should mention not only the literature that support a casual interaction between the two but also those that report that genome organization, in particular TADs domains are preexisting modules formed independently from the transcriptional state of the region that are interacting.

The manuscript needs extensive restructuring and systematic organization to clarify the messages and simplify the reading.

Major points are described below, but the authors should not limit only to those.

Introduction

The introduction paragraph is a bit too dense. I would suggest to describe better the concepts of the review, adding some sentences in order to guide the readers through what it will be explained in the body of the review.

Row 29 - 37: It is hard to understand what the authors want to say here. Imaging per se is not a method to link chromatin structure to function.

Optical imaging and Hi-C: Complementary and controversial views

What optical imaging is meaning?

The chapter is useful, but it is not describing systematically the topic; it jumps randomly from bulk HiC to single cell HiC to different FISH approaches. Also, distances should be properly treated; TADs contacts versus long range interactions, for instance. Transcription should not be included here, is out of the scope of the paragraph.

The figure 1 is not properly cited. It is also hard to understand what the Figure is showing; conventional HiC matrixes are not like those representing by the authors. The arrow is indicating the time, It is not clear how the time is influencing the process and how FISH versus

HiC is capturing the conformation of a given genomic locus.

Row 6 -7, page 6: Figure 2 is not showing chromosome organization, rather chromatin dynamics; also, it is not straightforward that the examples of probe labelling in the Figure 2 are referred to live imaging technologies (FISH, for instance); therefore, Figure 2 is not properly cited. More importantly, referred to figure 2 legend: it is really hard also in this case to understand which examples of labelling the authors are showing in the figure. The legend is too general.

Row 57-59, page 7: It is not clear that Figure 2 is showing what is written here.

Row 49-51, page 9: DNA is not a factor acting on transcription, please rephrase.

Row 22-24, page 10, Row 34 - 36, Page 14: This concept is still controversial; there are many findings suggesting that transcription and chromatin higher order structure are independent phenomena; in particular, 3 organization precedes transcription. This has to be properly commented.

Chromatin, transcription factor and polymerase II dynamics regulate transcription in space

This concept is too categorical. RNA pol II speed is very important for defining transcriptional trends. Not only space. Chromatin dynamics the same. I would rather include both TF and the rest in one chapter, both in time and space.

Reviewer 2

Comments to author:

The review addresses an important topic: the use of imaging technologies to study chromosome structure and dynamics and its relation to transcription. I have, however, several important concerns with how the literature is being portrayed in the review.

In In 51 of the introduction, Finn et al. (2019) is cited as providing Hi-C single cell evidence for heterogeneity but this paper used conventional FISH to show this. Also, earlier papers showing similar results should be cited as well (Cattoni, Nat. Comm 2017)

In In 54, " distances do not correlate to contact frequencies " in addition to Finn et al (2019) please also provide reference of Wang, (2016), Cattoni, et al (2017) as these were the first studies that compared directly the correlation between HiC and microscopy derived distance distributions.

In "Increased contact frequencies appear to correlate with spatial distances < 250nm rather than average distances". I do not understand what the authors mean.

Sequential FISH. The authors cite Beliveau et al (Ref 29) for this. But this reference should be quoted for the development of oligopaint, not sequential FISH. The first demonstration of sequential FISH for DNA was made in Wang et al (2016).

"the three-dimensional folding of chromatin regions was probed directly in single cells and in situ" authors cite Refs 30-33. I feel there is a bit of confusion here. The use of sequential FISH to detect chromatin conformation was first demonstrated by Wang (2016). A second wave of studies (Bintu, 2018 Ref 30; Cardozo, 2019 Ref 26; Mateo, 2019 Ref 27) showed that TADs can be also detected by sequential FISH. Nir et al (2018, Ref 31) used a quite different approach to resolve TADs at super-resolution using a variant of sequential FISH. Previous studies (Boettiger, 2016; Beliveau, 2015: Ref32; Szabo, 2018) used super-resolution to visualize TADs but this did NOT use sequential FISH. This should be written in a more clear fashion to avoid confusion and the order of the references should reflect the historical order of discoveries/inventions.

"Cooperative three-way interactions between loci [30]". I was not aware that Bintu reported this in their paper. I may be wrong, but it is worth double checking...

"to monitor the positioning of genes relative to nuclear compartments which can be visualized simultaneously [23,26-28]". The references cited here (at least 26 and 27) actually do not report on the position of genes relative to any compartment which are not even visualized!

"Imaging further showed that TADs appear as discrete nano-compartments, which are spatially arranged side by side [33,34], although contact frequencies between loci within TADs only slightly (~ 2-fold) increase compared to loci within neighboring TADs [35]." The first part of the statement should cite Boettiger, 2016 (first use of super-resolution to visualize TADs) now Wang et al (2016) which actually used sequential FISH to look at interactions between TADs. The second part of the statement (contact frequencies): this was reported first in Cattoni et al (2017), then by Finn (2018). Please avoid citing a News and Views in this context.

"Integrating multiplexed imaging of RNA [36] with sequential FISH". This paper did not really integrate multiplexed imaging of RNA and sequential DNA FISH. The first to do this integration was Mateo et al (2019). For multiplexed RNA imaging, authors should cite Chen Science et al (2015) in addition to Lubeck (2014).

"multiplexed DNA FISH imaging or reconstructed from Hi-C maps are currently limited to (i) fixed cells and (ii) a genomic resolution of ~ 1 kb [27]". Several problems here. Multiplexing DNA FISH refers to sequential FISH? I am confused here as multiplexing refers often to the use of combinatorial approaches which have not been reported for DNA. The genomic resolution reported by Mateo, I think, was 2kb not 1kb.

A major development was the simultaneous detection of transcriptional state and DNA architecture by sequential DNA-FISH. This was demonstrated in drosophila embryos (Mateo,

2019; Cardozo, 2019; Cardozo, 2020). As the authors start the introduction, this development is very important to be able to understand how transcription is related to DNA structure. However, this does not seem to be discussed anywhere??

By "transcription factor models" the authors mean 'transcription factory'? I am a bit confused here.s

"transcription has a large impact on chromatin structure and dynamics [64,65,71,140]". I don't understand the statement. Hug et al says in their abstract "the appearance of TAD boundaries is independent of transcription and requires the transcription factor Zelda for locus-specific TAD boundary insulation.". Therefore I don't understand why the authors quote this paper to support the claim that transcription impacts chromatin structure.

"Applying lattice illumination or random illumination..." what are the authors referring to? Lattice illumination? meaning Lattice light-sheet? What is random illumination?

I may have missed it, but the authors discuss extensively transcription factories but not enhancer clustering, which would provide a very different way of organizing transcription.

Super-enhancers and their potential roles in organizing transcription are not described either.

Roles of PRE clustering in repression of transcription does not seem to be described.

****Minor Comments****

"Models based on physics help interpret experimental data, but perturbing structure in situ to test causality between chromatin dynamics, folding and function is a challenging endeavour"

Not clear what the authors are trying to say. How do physical models perturb structure in situ?

Dear Andrew,

We thank the reviewers for their positive feedback on our review. We value their helpful comments which helped improve the manuscript.

The main point raised by reviewer 1 is that the manuscript needs restructuring, in particular of the introduction. We agree with him concerning the introduction, yet we believe that the body of the text reads coherently.

Below please find detailed answers to the points raised by both reviewers. We edited the text accordingly and highlighted changes in red.

Regarding the suggestion of listing different microscopy techniques, and then discussing each one, we feel that this would be redundant with a technical review I have published recently [<https://doi.org/10.1093/nar/gkaa135>]. We referred to this review in our introduction to help the reader to find a resource describing technical differences between imaging methods. The present review, however, focuses on how these imaging techniques enabled understanding of genome organization and dynamics with respect to transcription activity.

Finally, we edited the title to: *Navigating the crowds: Visualizing coordination between genome dynamics, structure and transcription*.

We believe that the review now nicely satisfies the standards of Genome Biology. The topic is more than ever of great interest to the community.

Best regards,
Haitham

Reviewer #1: General comments:

The topic is at the forefront, the point of view innovative and certainly of great interest. Unfortunately, this reviewer had hard time to understand what the review is describing. Chapters are often not systematically organized, and sometimes partially cover the literature of the field. The main problem is that the causality between transcription and genome organization is still an unresolved issue, and the review should mention not only the literature that support a causal interaction between the two but also those that report that genome organization, in particular TADs domains are preexisting modules formed independently from the transcriptional state of the region that are interacting.

We agree with the reviewer that the aim of the review was somewhat buried. We therefore extensively rewrote the introduction to make the overall aim, as well as the structure of the review, clear (in accordance with the reviewer's comment below).

We further agree with the reviewer that we did not mention studies arguing against a causal relationship between chromatin architecture/dynamics and transcription. Congruent with reviewer #2, we added a paragraph to highlight also these lines of evidence (see below).

The manuscript needs extensive restructuring and systematic organization to clarify the messages and simplify the reading.

Major points are described below, but the authors should not limit only to those.

Introduction

The introduction paragraph is a bit too dense. I would suggest to describe better the concepts of the review, adding some sentences in order to guide the readers through what it will be explained in the body of the review.

Row 29 - 37: It is hard to understand what the authors want to say here. Imaging per se is not a method to link chromatin structure to function.

We thank the reviewer for this helpful comment and suggestion, it is very important to clarify our aim from this review. So we re-wrote the this indicated paragraph with more details.

Optical imaging and Hi-C: Complementary and controversial views

What optical imaging is meaning?

We mean imaging based optical fluorescence microscopy. We changed it to **fluorescence imaging**.

The chapter is useful, but it is not describing systematically the topic; it jumps randomly from bulk HiC to single cell HiC to different FISH approaches. Also, distances should be properly treated; TADs contacts versus long range interactions, for instance. Transcription should not be included here, is out of the scope of the paragraph.

The figure 1 is not properly cited. It is also hard to understand what the Figure is showing; conventional HiC matrixes are not like those representing by the authors. The arrow is indicating the time, It is not clear how the time is influencing the process and how FISH versus HiC is capturing the conformation of a given genomic locus.

We briefly included the important development of the multiplexed detection of RNA with chromatin architecture, in accordance with Reviewer #2. Furthermore, we feel that these methods should be included in this paragraph since the review is focused on chromatin and interaction, while these methods beautifully combine both aspects. However, we agree that imaging-based approaches to interrogate transcription are better placed elsewhere. We thus rephrased the last paragraph of the section accordingly.

We included a clarifying paragraph at the end of the section to describe the message of Figure 1 in more detail (see below) and changed 'Hi-C-like data' to 'single cell Hi-C-like data' in the figure itself to clarify that the matrices are a representation of the polymers shown on the left. Furthermore, we edited the caption of the figure to better relate the illustrated polymer and matrices.

However, it is also evident that imaging specific sequences is necessary to place time-course data into functional context. In the future, labelling strategies will need to be designed for fluorescence live-cell imaging to simultaneously observe structural changes of chromatin and its functional consequences simultaneously in the same cell (Figure 1).

Row 6 -7, page 6: Figure 2 is not showing chromosome organization, rather chromatin dynamics; also, it is not straightforward that the examples of probe labelling in the Figure 2 are referred to live imaging technologies (FISH, for instance); therefore, Figure 2 is not properly cited. More importantly, referred to figure 2 legend: it is really hard also in this case to understand which examples of labelling the authors are showing in the figure. The legend is too general.

The reference to Figure 2 (originally at row 6-7, page 6) was moved to the end of the sentence since it is referring to the labelling of distinct genomic sides.

The purpose of Figure 2 is to illustrate different imaging modalities to interrogate the structural and/or dynamic behaviour of chromatin. We do not aim to highlight a few specific techniques (examples would be to label chromatin globally by a DNA stain or by expressing a fluorescent histone protein; single locus tagging can be done either via ANCHOR or CRISPR, among others). As such, we appreciate a rather general legend as we discuss advantages and disadvantages of the methods (wherever multiple exist) in the main text.

Row 57-59, page 7: It is not clear that Figure 2 is showing what is written here.

We agree that the reference to the figure is misplaced here. We removed the reference accordingly.

Row 49-51, page 9: DNA is not a factor acting on transcription, please rephrase.

We reformulated the sentence as follows:

Transcription is not only regulated in time, but depends largely also on the spatial organization of TFs and RNAP II [86]. Furthermore, transcription can modify the architecture and dynamics of DNA [87].

Row 22-24, page 10, Row 34 - 36, Page 14: This concept is still controversial; there are many findings suggesting that transcription and chromatin higher order structure are independent phenomena; in particular, 3 organization precedes transcription. This has to be properly commented.

We agree with the reviewer that this subject is a hot debate in the field. We therefore included the following paragraph in the Outlook:

However, the dependency between chromatin architecture and transcription regulation *in vivo* is a matter of debate. For instance, contacts between *cis*-regulatory elements precede gene activation during *Drosophila* development and remain regardless of the transcriptional state [161]. A few studies also found that enhancer-promoter proximity is not always indicative of the transcriptional state [85,162]. Strikingly, auxin-inducible degradation of the insulator protein CTCF, which acts together with cohesin in human cells to shape chromatin loop domains/TADs, has only a mild impact on transcriptional deregulation [163] and auxin-induced degradation of cohesin has a similarly modest effect on transcription [164]. However, several lines of research demonstrated that deletion of TAD boundaries can perturb transcription by interfering with ectopic enhancer-promoter contacts [165–167]. Genes which are deregulated upon loss of cohesin or CTCF are preferentially located at TAD boundaries, hinting toward a subtle regulatory role of factors shaping the genomic environment around *cis*-regulatory elements. Transient looping events by cohesin and/or TFs, which are likely not to be detectable by ensemble approaches such as Hi-C, could thus

constitute a flexible mechanism by which genes can be both activated and repressed, for instance as a response to stimuli.

Chromatin, transcription factor and polymerase II dynamics regulate transcription in space

This concept is too categorical. RNA pol II speed is very important for defining transcriptional trends. Not only space. Chromatin dynamics the same. I would rather include both TF and the rest in one chapter, both in time and space.

We agree with the reviewer and merged the two sections.

Reviewer #2:

The review addresses an important topic: the use of imaging technologies to study chromosome structure and dynamics and its relation to transcription. I have, however, several important concerns with how the literature is being portrayed in the review.

In ln 51 of the introduction, Finn et al. (2019) is cited as providing Hi-C single cell evidence for heterogeneity but this paper used conventional FISH to show this. Also, earlier papers showing similar results should be cited as well (Cattoni, Nat. Comm 2017)

We corrected this citation and added the suggested one.

In ln 54, "distances do not correlate to contact frequencies" in addition to Finn et al (2019) please also provide reference of Wang, (2016), Cattoni, et al (2017) as these were the first studies that compared directly the correlation between HiC and microscopy derived distance distributions.

We added the suggested references.

In "Increased contact frequencies appear to correlate with spatial distances < 250nm rather than average distances". I do not understand what the authors mean.

We rewrote this and the two following sentences to make the statement clear.

Sequential FISH. The authors cite Beliveau et al (Ref 29) for this. But this reference should be quoted for the development of oligopaint, not sequential FISH. The first demonstration of sequential FISH for DNA was made in Wang et al (2016).

We corrected the reference as suggested.

"the three-dimensional folding of chromatin regions was probed directly in single cells and in situ" authors cite Refs 30-33. I feel there is a bit of confusion here. The use of sequential FISH to detect chromatin conformation was first demonstrated by Wang (2016). A second wave of studies (Bintu, 2018 Ref 30; Cardozo, 2019 Ref 26; Mateo, 2019 Ref 27) showed that TADs can be also detected by sequential FISH. Nir et al (2018, Ref 31) used a quite different approach to resolve TADs at super-resolution using a variant of sequential FISH. Previous studies (Boettiger, 2016; Beliveau, 2015: Ref32; Szabo, 2018) used super-resolution to visualize TADs but this did NOT use sequential FISH. This should be written in a more clear fashion to avoid confusion and the order of the references should reflect the historical order of discoveries/inventions.

We thank the reviewer for pointing this out and adapted the references accordingly.

"Cooperative three-way interactions between loci [30]". I was not aware that Bintu reported this in their paper. I may be wrong, but it is worth double checking...

Bintu et al. (Science, 2018) could identify three-way interactions between loci as reported in Figure 5 of the publication.

"to monitor the positioning of genes relative to nuclear compartments which can be visualized simultaneously [23,26-28]". The references cited here (at least 26 and 27) actually do not report on the position of genes relative to any compartment which are not even visualized!

We corrected these citations.

"Imaging further showed that TADs appear as discrete nano-compartments, which are spatially arranged side by side [33,34], although contact frequencies between loci within TADs only slightly (~ 2-fold) increase compared to loci within neighboring TADs [35]." The first part of the statement should cite Boettiger, 2016 (first use of super-resolution to visualize TADs) now Wang et al (2016) which actually used sequential FISH to look at interactions between TADs. The second part of the statement (contact frequencies): this was reported first in Cattoni et al (2017), then by Finn (2018). Please avoid citing a News and Views in this context.

We added the suggested citations.

"Integrating multiplexed imaging of RNA [36] with sequential FISH". This paper did not really integrate multiplexed imaging of RNA and sequential DNA FISH. The first to do this integration was Mateo et al (2019). For multiplexed RNA imaging, authors should cite Chen Science et al (2015) in addition to Lubeck (2014).

We edited and added the suggested citations.

"multiplexed DNA FISH imaging or reconstructed from Hi-C maps are currently limited to (i) fixed cells and (ii) a genomic resolution of ~ 1 kb [27]". Several problems here. Multiplexing DNA FISH refers to sequential FISH? I am confused here as multiplexing refers often to the use of combinatorial approaches which have not been reported for DNA. The genomic resolution reported by Mateo, I think, was 2kb not 1kb.

We replaced multiplexed to sequential to make it clearer and more fit the context of DNA imaging. Also, we corrected the genomic resolution detected by Mateo to 2 kb.

A major development was the simultaneous detection of transcriptional state and DNA architecture by sequential DNA-FISH. This was demonstrated in drosophila embryos (Mateo, 2019; Cardozo, 2019; Cardozo, 2020). As the authors start the introduction, this development is very important to be able to understand how transcription is related to DNA structure. However, this does not seem to be discussed anywhere??

We agree that the developments from Mateo 2019 and Cardozo 2019 are of high importance and were briefly described in the section "Fluorescence imaging and Hi-C: Complementary and controversial views". However, we decided (also based on the comments of the other reviewer that brought up the valid argument that transcription in this section is out of scope) to keep the paragraph about these methods rather short.

By "transcription factor models" the authors mean 'transcription factory'? I am a bit confused here.

Indeed, we mean transcription factory model.

"transcription has a large impact on chromatin structure and dynamics [64,65,71,140]". I don't understand the statement. Hug et al says in their abstract "the appearance of TAD boundaries is independent of transcription and requires the transcription factor Zelda for locus-specific TAD boundary insulation.". Therefore I don't understand why the authors quote this paper to support the claim that transcription impacts chromatin structure.

We agree that Hug et al was not properly cited. Instead we included two other publications which support the view that transcription and chromatin impact each other.

"Applying lattice illumination or random illumination..." what are the authors referring to? Lattice illumination? meaning Lattice light-sheet? What is random illumination?

Here, we mean two different things; first, lattice light sheet illumination which could be in different lattice patterns, second, random (speckle) illumination. Random illumination microscopy, a super-resolved reconstruction of the sample, is formed numerically from several low-resolution images of the sample recorded under different uncontrolled speckle illuminations. To make it easy for the readers to differentiate between both illuminations, we edited the text to clarify both techniques and added the citation next to each one.

I may have missed it, but the authors discuss extensively transcription factories but not enhancer clustering, which would provide a very different way of organizing transcription.

Super-enhancers and their potential roles in organizing transcription are not described either.

Roles of PRE clustering in repression of transcription does not seem to be described.

Thanks for this suggestion, we added a paragraph about it as follows

Super-enhancers, clusters of putative enhancers in close proximity with unordinary high density of transcriptional co-activators binding [103,106,107], have shown the potential to form a nucleation point for LLPS, directed by master TFs [113]. The unusually high transcription activity of super-enhancers has to be 'contained' within insulated neighborhoods, demarcated by Hi-C-defined TAD boundaries, in order to specifically activate their target gene [113]. As such, the local chromatin architecture appears to be vital to prevent ectopic gene activation.

****Minor Comments****

"Models based on physics help interpret experimental data, but perturbing structure in situ to test causality between chromatin dynamics, folding and function is a challenging endeavour"

Not clear what the authors are trying to say. How do physical models perturb structure in situ? We rewrote the whole paragraph so now the message of this sentence is delivered in a clearer manner.

Note regarding Figure 3: The transcription factory model was mentioned during a talk by Clodagh C. O'Shea (Molecular and Cell Biology Laboratory, Salk Institute for Biological Studies) during the KITP conference 2020, which was held online (<http://online.kitp.ucsb.edu/online/chromosomes20/>). In particular, her team observed clusters of RNAP II which are in line with the transcription factory model. Unlike depicted in Figure 3, her data suggests that DNA is not excluded from transcription factories, but present in the interior of the transcription factory. We thus modified Figure 3 to incorporate this (unpublished) result.

Second round of review

Reviewer 1

The authors have satisfactorily improved the manuscripts, answering to the majority of my concerns.

However, still I believe that Figure 1 is not clear and misleading, representing imaging data with the matrix of HiC data.

Reviewer 2

There are still some outstanding issues that need attention.

citation issues

- There are many reviews recently published on the use of imaging technologies to address chromosome structure. These are not cited properly.

lack of clarity

- "time-resolved whole-chromatin imaging recently revealed the spatio-temporal correlation of chromatin dynamics at nanoscale resolution" what is meant here??
Correlation of what with what?

- is the review about live cell imaging only or not? if the former is true then focus exclusively on this. Otherwise, be comprehensive.

- "Hi-C and derived methods can only assess the probability of crosslinking which is a measure of the

81 frequency that chromatin loci are in proximity, probing spatial distances (< 250 nm)." I don't think this has been demonstrated. In any case, no reference is provided.

- Regarding random-illumination. I miss any explanation of differences between the methods. But in fact, I don't think random illumination should be cited in absence of a publication (not a bioRxiv) documenting the capabilities of this method and a proper comparison with existing methods. Putting these two methods side by side is not advisable, given that one is well-established (LLS) while the other is not published yet.

Dear Andrew,

We are happy to submit a revised version of our manuscript "Navigating the crowds: Visualizing coordination between genome dynamics, structure and transcription", which we hope is now suitable for publication in genome Biology. We have addressed the remaining comments by the reviewer 2 and the inquiry from reviewer 1, as detailed in our 2nd Response to Reviews document, and have marked the corresponding changes in the combined document in red.

Best regards,
Haitham Shaban

Reviewer #1: The authors have satisfactorily improved the manuscripts, answering to the majority of my concerns.

However, still I believe that Figure 1 is not clear and misleading, representing imaging data with the matrix of HiC data.

The representation of distance (in contrast to contact frequency) for imaging data is quite common for figures displaying the 3D chromatin configuration from imaging data (see e.g. Bintu et al., 2018, Science; Mateo et al., 2019, Nature; Su et al., 2020, Cell). Furthermore, we clearly label which matrix corresponds to Hi-C-like and which to imaging-like data and note that either Hi-C counts OR spatial distance is shown. Yet we agree that readers which are unaware of these publications may be helped by some explanation on what the matrices represent. We thus explained the rationale behind the matrix representation in more detail in the caption:

The illustrative maps were created by tracing the contour of the polymer shown on the left and computing the pairwise distance between any two loci which is shown in the imaging-like matrix. The Hi-C map is a thresholded version of the distance map and shows contacts only at small spatial distances. Yet note that there may be a broad distance distribution underlying measured Hi-C contacts [176] and as such the illustration is highly simplified.

Reviewer #2: There are still some outstanding issues that need attention.

citation issues

- There are many reviews recently published on the use of imaging technologies to address chromosome structure. These are not cited properly.

We agree that some important recent reviews on the topic were not cited originally. We included (some of) those in references 8 - 11:

For a comprehensive comparison between different imaging techniques used to studying genome organization and transcription, we refer the reader to recent reviews [8–11].

lack of clarity

- "time-resolved whole-chromatin imaging recently revealed the spatio-temporal

55 correlation of chromatin dynamics at nanoscale resolution" what is meant here?? Correlation of what with what?

We clarified what 'correlation' points to in a revised version of the sentence and gave a little more background information:

While super resolution imaging-based approaches are often performed in fixed cells, time-resolved whole-chromatin imaging recently revealed that chromatin moves in a spatially and temporally

correlated manner at nanoscale resolution [10,11]. Such correlated motion might be caused by active mechanisms [12], including transcription [13,14], and/or polymer properties [15].

- is the review about live cell imaging only or not? if the former is true then focus exclusively on this. Otherwise, be comprehensive.

The review's main topic is live cell imaging and what it can teach us about chromatin structure, dynamics and their relation to transcription. However, our knowledge heavily relied on sequencing-based and fixed-cell-based methods. It is thus important to lay out what advantages and disadvantages these methods have and where live-cell imaging can contribute knowledge beyond other methods. Therefore, we decided to (briefly) include Hi-C and related methods as well as fixed cell imaging in the review. Yet the brevity does not allow to discuss all methods in depth but should point out some main insights and main limitations of other methods to study chromatin. We make this point of view clearer in the abstract (and already in the introduction):

Here, we describe how recent advances in quantitative imaging techniques **overcome caveats of sequencing-based (Hi-C and related) methods by enabling** direct visualization of transcription factors and chromatin at high resolution, at the scale of single genes to that of the whole nucleus.

- "Hi-C and derived methods can only assess the probability of crosslinking which is a measure of the 81 frequency that chromatin loci are in proximity, probing spatial distances (< 250 nm)." I don't think this has been demonstrated. In any case, no reference is provided.

These values have been recently revisited during the revision of the manuscript. The authors in Su *et al.*, 2020, Cell found that ensemble imaging- and Hi-C-based contact matrices are most similar when contacts are assumed at a distance cutoff around 400-600 nm (they chose 500 nm). We thus adapted the stated values accordingly and give the reference publication.

Hi-C and derived methods can assess the probability of crosslinking which is a measure of the frequency that chromatin loci are in proximity, probing spatial distances in the range ~ 400 – 600 nm [29].

- Regarding random-illumination. I miss any explanation of differences between the methods. But in fact, I don't think random illumination should be cited in absence of a publication (not a bioRxiv) documenting the capabilities of this method and a proper comparison with existing methods. Putting these two methods side by side is not advisable, given that one is well-established (LLS) while the other is not published yet.

We deleted this citation for avoiding any confusion